# Prevalence of Aspergillus-Derived Mycotoxins (Ochratoxin, Aflatoxin, and Gliotoxin) and Their Distribution in the Urinalysis of ME/CFS Patients

**DOI:** 10.3390/ijerph19042052

**Published:** 2022-02-12

**Authors:** Ting Yu Wu, Taura Khorramshahi, Lindsey A. Taylor, Nikita S. Bansal, Betsy Rodriguez, Irma R. Rey

**Affiliations:** 1Dr. Kiran C. Patel College of Osteopathic Medicine, Nova Southeastern University, Fort Lauderdale, FL 33328, USA; tw1184@mynsu.nova.edu (T.Y.W.); tk452@mynsu.nova.edu (T.K.); lt1010@mynsu.nova.edu (L.A.T.); nbansal2016@fau.edu (N.S.B.); betsy.rodriguez@hmhn.org (B.R.); 2Institute of Neuro-Immune Medicine, Nova Southeastern University, Fort Lauderdale, FL 33328, USA

**Keywords:** ochratoxin A, aflatoxin, gliotoxin, myalgic encephalomyelitis, chronic fatigue syndrome, urinalysis

## Abstract

Myalgic encephalomyelitis/chronic fatigue syndrome (ME/CFS) is a known complex, multi-organ system disorder with a sudden or subacute onset. ME/CFS occurs most commonly among women between 30 and 50 years of age. The current diagnostic criteria of ME/CFS, as defined by the Centers for Disease Control and Prevention, includes: profound fatigue and post-exertional malaise (>6 mo) unrelieved by rest, persistent cognitive impairment or orthostatic intolerance, and chronic unrefreshing sleep. Despite reported associations between ME/CFS onset and exposure to infectious agents (viral, bacterial, or fungal), the pathophysiology of ME/CFS remains unknown. In this prevalence study, we investigated the rates of Aspergillus-derived toxin levels, Aflatoxin (AF), Ochratoxin A (OTA), and Gliotoxin (GT), in the urinalysis of 236 ME/CFS patients with a history of chronic exposure to mold (i.e., from water-damaged buildings). Among ME/CFS patients reporting chronic exposure to mold, we found evidence of exposure in 92.4 percent of patients, with OTA being the most prevalent mycotoxin. Mold distributions (OTA, AF, and GT) in the urinalysis all demonstrated right skewness, while the distribution of age of ME/CFS patients diagnosed showed no deviation from normality. This study aims to provide preliminary, epidemiological evidence among ME/CFS patients who were diagnosed in South Florida with a history of exposure to mycotoxins. Based on these findings, we proposed how future control studies should approach investigating the association between chronic mold exposure and the diagnosis of ME/CFS.

## 1. Introduction

Myalgic encephalomyelitis/chronic fatigue syndrome (ME/CFS) is a chronic, complex, and debilitating disease, impacting up to three percent of the United States population. The condition disproportionately affects more women than men [1] and often follows an episode of viral or bacterial infection [2]. Once diagnosed, an estimated quarter of patients become bedbound or housebound, and many are unable to work, attend school, or perform daily tasks after 1–2 years [3,4].

Currently, diagnosis of ME/CFS is based on presenting symptomatology, and uses a process of elimination for diagnosis, especially since patients often struggle with diffuse, intermittent symptoms over an extended period of time [5]. The most recent diagnostic criteria of ME/CFS as defined by the Centers for Disease Control and Prevention (CDC) includes persistent cognitive impairment or orthostatic intolerance, in addition to at least one of the three following symptoms: (1) a substantial reduction in pre-illness activities with profound fatigue lasting more than six months, which is not caused by ongoing excessive exertion and is unrelieved by rest; (2) post-exertional malaise (PEM); or (3) frequent and severe unrefreshing sleep [6]. The treatment for ME/CFS also relies on presenting symptomatology, since, despite fruitful efforts, the underlying mechanisms of disease activity have yet to be fully elucidated.

While the etiology of ME/CFS is not fully known, several recent studies have highlighted disruptions in neurological, immune, and endocrine function in patients with ME/CFS [7,8,9]. The onset of disease typically follows an episode of severe viral or bacterial insult and its progression is linked to active inflammatory responses that disrupt immune recovery [10]. Chronic inflammation caused by having both a suppressed immune system and active infections therefore is thought to compound the severity of ME/CFS [11].

Researchers have identified neuroendocrine abnormalities that result in cognitive impairment, decline in working memory, sleep dysfunction, PEM, and concomitant psychiatric disorders [12]. In addition, metabolic abnormalities are often seen among ME/CFS patients. These include significant microbiome changes, mitochondrial dysfunction, and peroxisomal dysregulation [9]. Several previous studies have also observed that the presence of mold exposure, particularly Aspergillus toxins (i.e., Aflatoxin (AF), Ochratoxin (OTA), and Gliotoxin (GT)), may serve to propagate disease progression [8,9]. As such, further insight into the impacts of mold exposure on ME/CFS symptomatology, pathophysiology, progression, and disability is important, as ME/CFS is now understood to be a chronic, progressive, and multi-factorial disease.

### 1.1. Aflatoxin (AF)

AFs have long been considered by the World Health Organization as some of the most poisonous toxins produced by mold. They proliferate in contaminated staple crops, such as corn, wheat, sorghum, and rice cereals, which makes them a high-priority concern for food health and safety worldwide. Acute toxicity of large amounts of AFs, known clinically as aflatoxicosis, can cause severe poisoning and death [13]. AFs have the ability to cause damage to the genome by DNA intercalation and hindering DNA repair, which can lead to p53 instability and carcinogenic mutations [14,15]. AFs accumulate mostly inside the liver, where these toxins are detoxified through the CYP system. The accumulation of aflatoxins in the liver has been shown to lead to hepatic DNA damage via lipid peroxidation and oxidative stress [16,17]. Due to their ability to affect the genome, aflatoxins can cause severe toxicity when they are not properly eliminated from food products [18].

### 1.2. Ochratoxin A (OTA)

OTA is a naturally occurring foodborne mycotoxin that not only infects cereal grains, but can also contaminate a variety of agricultural products, including corn, wheat, dried fruits, wine, and coffee beans [19,20,21]. Among the types of ochratoxins, ochratoxin A has been studied the most. OTA is thought to cause genome damage by inducing DNA adducts and quinone formation in the nucleus through oxidation [22,23]. Although it is unclear how the exact mechanism of carcinogenesis occurs, OTA has been also shown to increase oxidative stress in the mitochondria with known deleterious effects on DNA function, lipid peroxidation, cell anabolic metabolism, and cellular respiration [22]. Its carcinogenicity is most notable in the kidney, where OTA has been shown to cause permanent nephropathies and urinary tract tumors [22,24]. Interestingly, OTA has been cited as a potential cause of Balkan endemic nephropathy (BEN), a chronic tubulointerstitial disease endemic to countries in Southeastern Europe with the potential to cause urothelial malignancy; however, its role in BEN is still debated [22,25,26].

### 1.3. Gliotoxin (GT)

GT is found in compost piles, decaying vegetation, and water-damaged buildings. It is a potent mycotoxin that is produced by Aspergillus fumigatus and is the leading cause of invasive aspergillosis worldwide. Among other mycotoxins, GT has the widest range of effects on the immune system, including the disruption of dendritic antigen-presenting processes, activating ROS-mediated apoptosis, and disrupting the integrity of the epithelial and endothelial barriers to enhance systemic fungal invasion [27,28]. GT causes immunosuppression in the body by blocking the activation of NF-kB (Nuclear Factor kappa-light-chain-enhancer of activated B-cells) and inhibiting the formation of perforins and Fas-Fas ligand complexes induced by cytotoxic T-lymphocytes (CTLs) in target cells [29]. Because they cause immunosuppression, these mycotoxins can cause systemic manifestations, ranging from eye irritation to respiratory arrest, in addition to more general and nonspecific symptoms, such as headaches, malaise, skin hypersensitivity, and alterations to olfactory and taste sensations [30].

### 1.4. Purpose of Study

Studies on the environmental toxicology of ME/CFS (i.e., communicable viruses, bacteria, and household mold etc.) have demonstrated that a high prevalence of chronic mycotoxin in the body may contribute to the presenting symptomatology and immune-compromised state of ME/CFS patients [31]. Clinically, managing symptoms of ME/CFS has included administering glutathione, with patients reporting a significant reduction in PEM symptoms [32]. A thorough understanding of the different epidemiological factors that can affect ME/CFS patients susceptible to environmental toxins is needed in order to establish a foundation on which future control studies can examine disease outcome and prognosis. The aim of this preliminary study is to establish a detailed prevalence analysis of ME/CFS patients by age, gender, and mycotoxin type (OTA, AF, and GT) and their distributions in a sample of 236 ME/CFS patients, all of whom reported a history of exposure to water-damaged buildings throughout their lifetime.

## 2. Materials and Methods

### 2.1. ME/CFS Patients

In our prevalence study, a total of 236 ME/CFS patients were recruited for urinalysis to determine their exposure to three common types of mold toxins (OTA, AF, and GT). ME/CFS patients were seen by a clinician at Nova Southeastern University’s (NSU) Institute of Neuro-Immune Medicine (INIM) clinic in Fort Lauderdale, FL, and were recruited from August 2019 to February 2021 based on a history of having occupied buildings that predisposed them to chronic exposure to mold and mycotoxins (i.e., water-damaged buildings). The patients included in this study met the following inclusion criteria: diagnosed with ME/CFS, treated within the INIM clinic, and had a concurrent history of mold exposure prior to glutathione and charcoal treatment. Only patients who had medical insurance sufficient to cover the cost of analysis and subsequent testing, or were able to pay out-of-pocket, were recruited for this study in an effort to prevent additional out-of-pocket costs. Levels of mycotoxins (OTA, AF, and GT) were compiled from urinalysis reports and stratified by gender, age, and mycotoxin type to determine the prevalence of chronic exposure in each category. Lab samples were analyzed by RealTime Laboratories or The Great Plains Laboratory (GPL) based on patient preference and/or insurance coverage.

### 2.2. Mycotoxin Testing from RealTime Laboratories

Mycotoxin testing from RealTime Laboratories was conducted using the enzyme-linked immunosorbent assay (ELISA) technique with antigen–antibody reactions to detect the presence of specific mycotoxins in the urine samples.

### 2.3. Mycotoxin Testing from The Great Plains Laboratory

Urinalysis of mycotoxins from GPL was conducted using a technique that combined advanced mass spectrometry with liquid chromatography. In addition, a creatinine correction was applied to the urinalysis to minimize variations across samples for differences in total body water, fluid intake, and urine osmolality.

## 3. Results

### 3.1. Prevalence of Mycotoxins in Urinalysis

The overall prevalence of at least one mycotoxin exposure (OTA, AF, or GT) among a sample of 236 ME/CFS patients seen at the NSU’s INIM was 92.4 percent (Figure 1). Specifically, among patients who were diagnosed in South Florida, the prevalence of OTA was 80.9%, and the prevalence of AF and GT was 29.6% and 39.8%, respectively (Figure 2). The portion of patients who tested positive for all three types of mycotoxins in their urine was 54.2% (Figure 1). These findings reflect the geography of ME/CFS patients in this study; varying regions may differ in their detection of mycotoxin types. The average prevalence of at least one mycotoxin in females (*n* = 182) was 91.8% (OTA = 0.813, AF = 0.308, and GT = 0.379), and the average prevalence among males (*n* = 54) was 94.4% (OTA = 0.796, AF = 0.259, and GT = 0.463) (Figure 3a, Table 1). The most prevalent toxin in both female and male sampling populations was OTA. The least prevalent in both female and male sampling populations was AF. Despite large differences in the sample size of female versus male patients, the prevalence of all three mycotoxin types was graphically comparable between both male and female patients, where levels of OTA > GT > AF. In terms of age, data were stratified by age over and under 65 years of age to determine the prevalence pattern of mycotoxin type in patients under the age of 65 (*n* = 187), as compared with patients over 65 years of age (*n* = 49). For patients under 65 years of age, the overall prevalence of at least one mycotoxin was 93 percent (OTA = 0.665, AF = 0.182, GT = 0.262) (Table 1). Above 65 years of age, the overall prevalence of at least one mycotoxin was 89.8 percent (OTA = 0.144, AF = 0.114, GT = 0.136) (Table 1). Both age subgroups showed graphically similar patterns of prevalence, with OTA > GT > AF (Figure 3b). A higher prevalence of mycotoxins was observed in patients diagnosed at an earlier age (<65 yo) than those diagnosed later (>65 yo), because the majority of patients diagnosed at a younger age also reported a longer duration of exposure to mycotoxins prior to the onset of ME/CFS. Data for Trichothecenes mold were also collected, but for a majority of patients, they fell below detection and were excluded from our analysis of mycotoxin-type exposure prevalence.

### 3.2. Mycotoxin Distributions in Urinalysis (N = 236) and the Test for Normality

A histogram was plotted. The data showed mean OTA levels equal to 12.371 ± 12.183 ppb (Figure 4). OTA levels in the urinalysis revealed a right-skewed distribution, by which a normal quantile-quantile plot was made and a Kolmogorov-Smirnov normality test conducted (*p* < 0.001) to confirm statistical deviation from normal distribution (Figure 5 and Figure 6). The same analyses were conducted for AF and GT, respectively. Mean AF levels were 0.787 ± 1.717 ppb. Mean GT levels were 126.063 ± 584.604 ppb. Both AF and GT distributions also showed similar skewing and a deviation from normality (*p* < 0.001). These findings suggest that, for statistical analyses, comparing mean differences of mold toxin levels between patients and controls (i.e., paired *t*-test), data from mold distribution in the urinalysis may require a logarithmic transformation or recruiting a larger sample size to employ the central limit theorem (see discussion).

### 3.3. Age Distribution and Normality in ME/CFS Patient Sample

ME/CFS is a chronic disease with symptoms that may persist in many patients despite treatment. Based on the importance of disease prognosis, we determined that an analysis of age distribution in a sample of 236 ME/CFS patients was needed to capture the population’s profile.

In this group of patients, the mean age of ME/CFS patients was 53 years old (Figure 7). Age distribution reflected a spike in prevalence at approximately 40 years of age and a bi-modality of between 30 and 50 years of age, which is consistent with other former studies [33]. The distribution of age at diagnosis of 236 ME/CFS patients seeking treatment at NSU’s INIM in South Florida depicted a statistically significant (*p* = 0.007) normality fit using Kolmogorov-Smirnov normality testing (Figure 8 and Figure 9). As such, we can infer that analysis of age, as a variable, is amenable to different types of statistical testing in the ME/CFS population.

## 4. Discussion

ME/CFS is not only a burden on the patient’s overall health, physical functioning, mental capacity, and emotional stability, but it is also a direct burden on the healthcare system and economy, with an irreversible loss of productivity of approximately USD 17 to 24 billion annually [34]. While the underlying mechanisms of disease activity have yet to be fully understood, an increasing number of research studies in environmental health have highlighted the role of infectious toxins in the prognosis of ME/CFS [10,35,36,37]. Specifically, research findings have demonstrated a high prevalence of mycotoxins among ME/CFS patients [2,38], which are likely contributing to the progression of disease pathophysiology. Repeated exposure to mold toxins increases nephrotoxicity, which makes these toxins harder to eliminate from the kidneys and may be the reason behind how they cause sustained inflammation and chronic immune dysfunction in the body. Mycotoxins are also known to damage the immune system directly by disrupting CD4 to CD8 T cell-count ratios and lowering natural killer cell reserves [39]. Long-term exposure to carcinogenic mold agents in the environment is associated with symptomatic debilitation, leading to the progressive, additive decline of the body’s immune health [2]. Still, current diagnosis and management of ME/CFS relies on addressing presenting symptomatology. As such, understanding the role of mycotoxins in ME/CFS development may help to improve management and overall patient outcomes.

Mycotoxicosis, defined as the accumulation of fungal toxins in the body, has been shown to be associated with carcinogenic, teratogenic, neurotoxic, and immuno-toxic effects [30,40,41]. These toxins are found in contaminated food sources, and when ingested regularly, they can infect the gut epithelium and perturb the integrity of the gut lining and gut microbiota over time [42]. Some studies have also reported that a few mycotoxins, including aflatoxins, can be transported from olfactory neurons into the temporal lobe and cause encephalopathy [43]. Once these toxins have entered the human body, they can be extracted from urine, sputum, or tissue sample and detected with immunochemistry [44].

Mold toxins, particularly Aspergillus toxins (i.e., AF, OTA, and GT), proliferate in warm, humid conditions, such as in food, crops, dust, or damp indoor ventilation systems [43,45,46]. The total estimate of mold contamination stated by the United States Department of Agriculture was suspected to be around 25 percent in 1985, which is most likely an underestimation of the true magnitude of contamination in all transnational staple crops today [47]. Other environmental studies have looked at how climate change affects the growth rate of mold toxins worldwide in food staples and damp buildings [48]. These findings together warrant environmental policies to ensure food safety and prevent the contamination of agricultural staples worldwide. Because mycotoxins have several modes of transmission, for example through the ingestion of food products or inhalation from indoor air [45], they can cause insidious infections in immunocompromised patients if they are not regularly eliminated following sustained exposure.

Results from this study demonstrate a high overall prevalence of mycotoxins in the urinalysis of ME/CFS patients who have an exposure history to mycotoxins. Specifically, our findings on prevalence by age and gender were consistent with results published by other researchers [33]. Further analysis of mycotoxin type (OTA, AF, and GT) demonstrated right skewness with a deviation from normality. A limitation of this study was that we only recruited patients who were exposed to water-damaged buildings and those who could cover the costs of urinalysis testing through insurance or self-pay. As a result, our study provided preliminary evidence on the prevalence of various epidemiological factors (including age, gender, and mycotoxin type) affecting ME/CFS patients who were diagnosed in South Florida, without the recruitment of control groups. Future studies should look to reduce sampling bias by broadening their inclusion criteria and recruiting a larger sample size of exposed patients.

A clinical study conducted previously by Brewer et al. in 2013 compared levels of OTA and AF in the urinalysis of 107 ME/CFS patients to healthy controls residing in Kansas City [49]. Our preliminary findings found similar results for the prevalence of OTA and AF, but we were unable to determine their statistical significance without the addition of controls. Our analysis suggests that, due to right-skewed distributions of OTA, AF, and GT, larger sample sizes (*n* > 29) are needed to employ the central limit theorem and conduct statistical testing of differences in mean mycotoxin levels in the urinalysis of exposed ME/CFS patients as compared to healthy controls. For control studies with smaller sampling sizes (*n* < 29), we recommend a proportion analysis using a two-sample *t*-test or Fischer’s exact test to assess the probability of testing positive for mycotoxins in urine, as opposed to differences in mean mycotoxin levels in the urinalysis. Findings from this preliminary prevalence study may help establish variables (i.e., age, gender, and mycotoxin type) for future epidemiological research to assess their strength of association to the diagnosis of ME/CFS in patients with a known, chronic exposure to mold.

## 5. Conclusions

Findings from our preliminary study revealed that ME/CFS patients who reported a history of mycotoxin exposure had evidence of exposure to OTA, AF, and GT in their urinalysis. Distributions of these Aspergillus-derived mycotoxin types showed right-skewed deviations from normality, suggesting the need for subsequent control studies with larger sampling sizes to conduct appropriate statistical analyses. It remains unclear, however, whether these exposed ME/CFS patients constitute a unique subgroup of ME/CFS patients with a high incidence of detectable mycotoxins, or whether mold toxins contribute to the overall onset or progression of the disease in all patients reporting a history of chronic exposure to mold. Based on these findings, future studies are warranted to understand the extent that mycotoxin exposure contributes to ME/CFS onset, progression, and prognosis. In addition, expanding to a more diverse patient population with the analysis of additional environmental toxins and assessing the relationship between mold exposure and severity of ME/CFS are equally justified.

## Figures and Tables

**Figure 1 ijerph-19-02052-f001:**
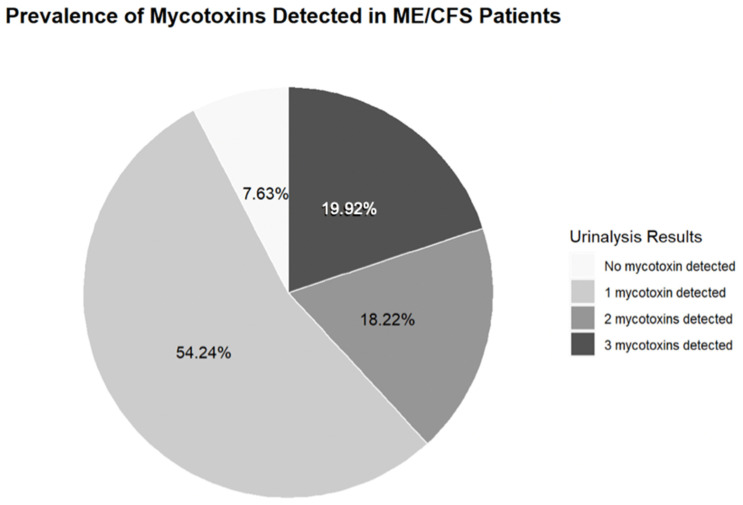
Among ME/CFS patients (N = 236) with a chronic exposure history to mold, the percentage of patients who did not test positive for mycotoxins at all was 7.63%. The proportion of those who tested positive for only one mycotoxin, regardless of type, was 54.24%. Percentage of patients who tested positive for all three types of mycotoxins was 19.92 percent. The overall prevalence of mycotoxins in the urinalysis was 92.4 percent.

**Figure 2 ijerph-19-02052-f002:**
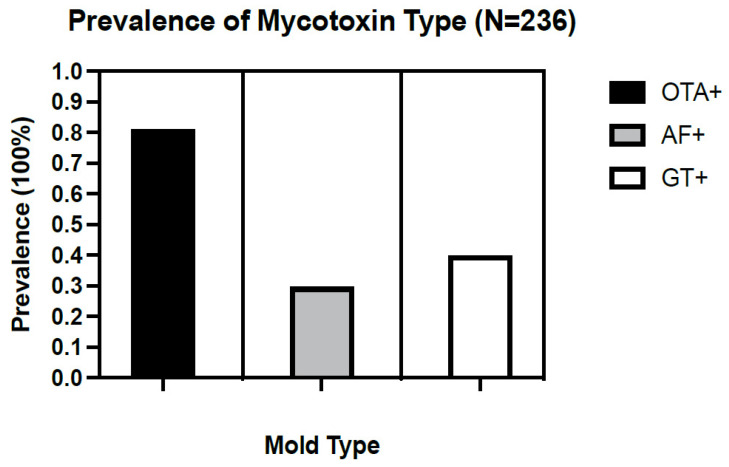
Prevalence of mycotoxin types in the urinalysis of ME/CFS patients (N = 236) who were diagnosed in South Florida and treated at INIM with a reported environmental history of chronic exposure to mold showed prevalence of Ochratoxin A was 80.9%, Aflatoxin was 29.6%, and Gliotoxin was 39.8%.

**Figure 3 ijerph-19-02052-f003:**
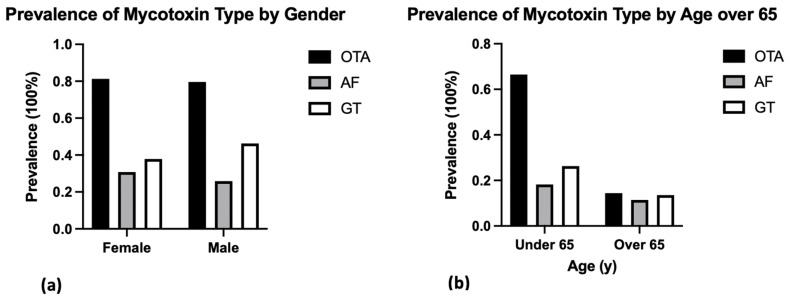
(**a**) The overall prevalence of at least one exposure in females (*n* = 182) was 91.8%, and in males (*n* = 54) was 94.4%. In females, OTA was 81.3%, AF was 30.8%, and GT was 37.9%; in males, OTA was 79.6%, AF was 25.9%, and GT was 46.3%. OTA was the most prevalent mycotoxin in both genders. (**b**) Among patients under 65 years of age (*n* = 187), the overall prevalence of at least one exposure was 93%. Under 65 yo, the prevalence of OTA was 66.5%, AF was 18.2%, and GT was 26.2%. For patients over 65 years of age (*n* = 49), the overall prevalence of at least one exposure was 89.8 percent. Over 65 yo, the prevalence of OTA was 14.4 percent, AF was 11.4 percent, and GT was 13.6 percent.

**Figure 4 ijerph-19-02052-f004:**
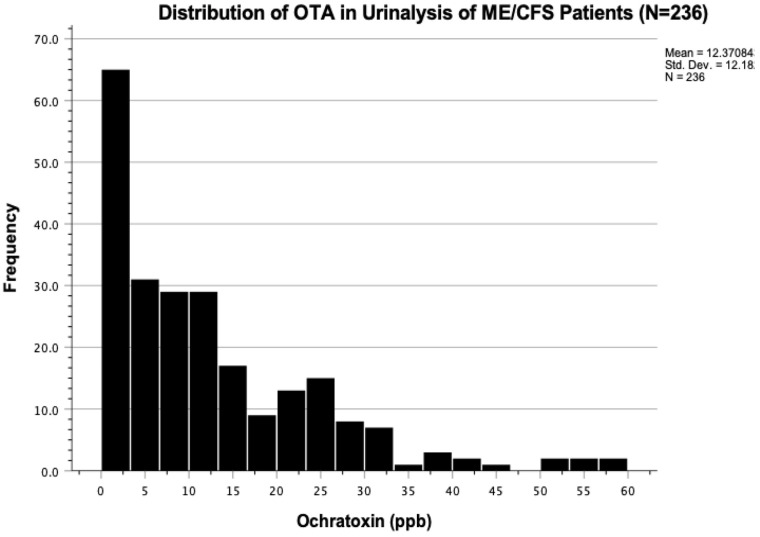
Ochratoxin A in urinalysis of 236 ME/CFS patients with chronic exposure to mold showed a right-skewed distribution with mean OTA levels of 12.37 ± 12.18 ppb.

**Figure 5 ijerph-19-02052-f005:**
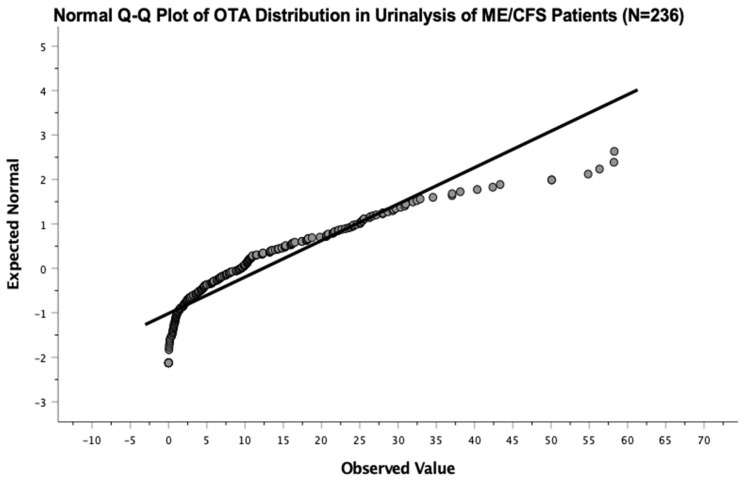
Normal quantile-quantile plot of OTA distribution in ME/CFS patients showed deviation from normality.

**Figure 6 ijerph-19-02052-f006:**
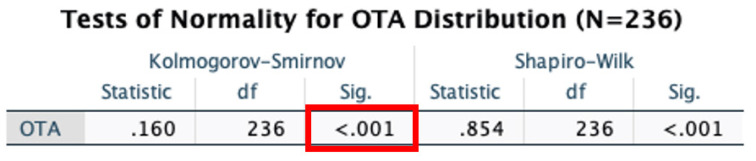
Kolmogorov-Smirnov normality test showed no significance (*p* < 0.001) for fit of normality of OTA distribution in urinalysis of ME/CFS patients.

**Figure 7 ijerph-19-02052-f007:**
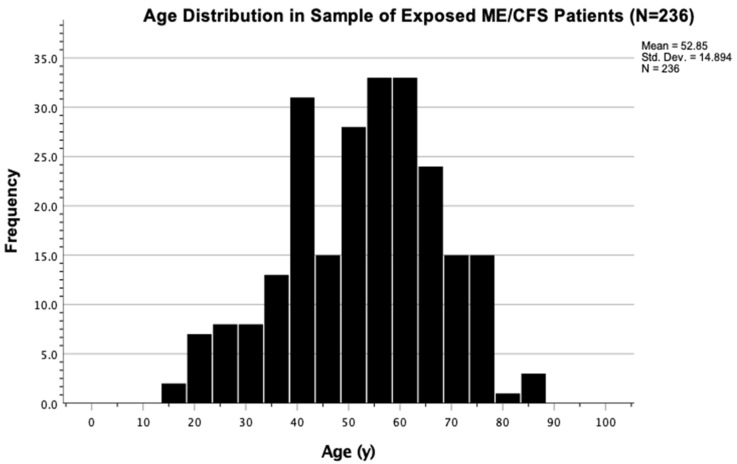
The mean age at diagnosis in ME/CFS patients who reported a history of chronic mold exposure was determined to be close to 53 years old. Bimodal peak incidence of age was noted at 40 years and 55–65 years of age.

**Figure 8 ijerph-19-02052-f008:**
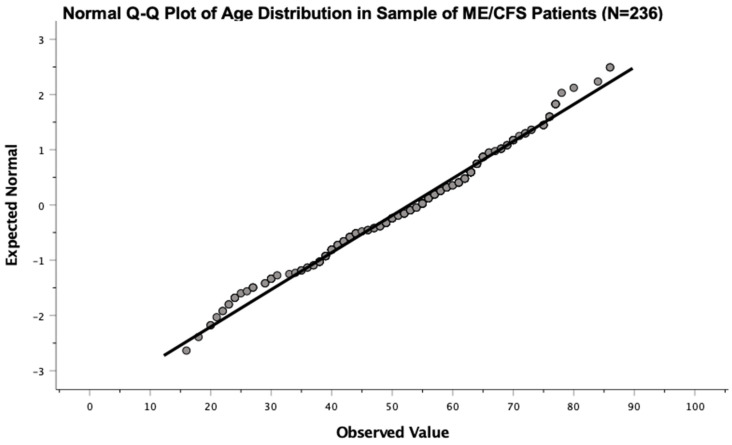
Normal quantile-quantile plot of age distribution in ME/CFS patients showed normality fit.

**Figure 9 ijerph-19-02052-f009:**
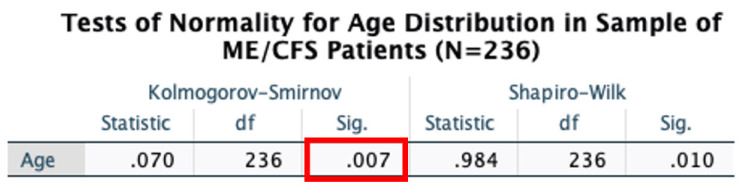
Kolmogorov-Smirnov normality test showed a statistically significant fit of normality (*p* = 0.007) for the distribution of age in ME/CFS patients with a history of chronic exposure to mold.

**Table 1 ijerph-19-02052-t001:** Number of ME/CFS patients who tested positive for mycotoxins in urinalysis by gender and age.

Prevalence of Mycotoxins	OTA	AF	GT	At Least One Mycotoxin
Gender (F = 182, M = 54)	F = 148 (81.3%)	F = 56 (30.8%)	F = 69 (37.9%)	F = 167 (91.8%)
	M = 43 (79.6%)	M = 14 (25.9%)	M = 25 (46.3%)	M = 51 (94.4%)
Age (Under 65 y = 187, Over 65 y = 49)	<65 y = 157 (66.5%)	<65 y = 43 (18.2%)	<65 y = 62 (26.2%)	<65 y = 174 (93.0%)
	>65 y = 34 (14.4%)	>65 y = 27 (11.4%)	>65 y = 32 (13.6%)	>65 y = 44 (89.8%)

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
