# Peer review of "Prevalence of Aspergillus-Derived Mycotoxins (Ochratoxin, Aflatoxin, and Gliotoxin) and Their Distribution in the Urinalysis of ME/CFS Patients"

_ijerph, 2022, doi:10.3390/ijerph19042052_

Round 1
Reviewer 1 Report
See attached file: Mycotoxin in MECFS Peer Review

Author Response
Dear Review 1, thank you for taking the time to provide us your feedback line by line. They were exceptionally helpful to our revisions. Please see attachment.
Thank you, again.

Reviewer 2 Report
The current study is a report on the prevalence of Aspergillus-derive mycotoxins in CFS patients. Wu et.al. focus on the presence of 3 primary mycotoxins – ochratoxin, aflatoxin, and gliotoxin in the patients. They report that ochratoxin is one of the most prevalent toxins in CFS patients with mold exposure. Further, the authors stratify patients with age and gender. The distribution of the three toxins was similar in males and females – OTA>GT>AF and they also identify the mean age at diagnosis to be 56 years of age. This study aims to provide preliminary evidence for a detailed analysis of the presence of mycotoxin in CFS patients stratified by age, gender, and type of mycotoxin. Given the lack of specific cause and treatment of CFS, studies investigating the cause of this disorder are important.
While I understand the need for the analysis and the need to identify parameters of comparison, I am struggling to find the unique nature of this study. A study in 2013 by Brewer et.al. performed a similar analysis and was met with some valid criticism regarding their control selection. However, their findings were very similar. Pertaining to this study, I am confused about the control groups. I do not see any mention of control groups. To mention that mycotoxins have a significant effect on CFS we need to see whether patients exposed to mold who do not present with CFS have lower levels of mycotoxins. There is no control group here. I do think that the aim of this study is important as it adds useful information regarding, the type of analysis that is necessary, and the sample size needed. However, without a control group can we really say that levels of mycotoxin are critical. If there are technical difficulties in getting the necessary control samples, it needs to be addressed and the inference of the results modified accordingly. Further, highlighting the novelty of this study in detail for readers who are not familiar with the disease would broaden the reader pool and help communicate the importance of this study.
Overall, I think the aim of this study is important and the data is interesting but a further justification of the novelty and need of this study will help the reader.
Author Response
Dear Review 2, thank you for providing us your invaluable feedback and insight into our manuscript. We have addressed all your points in this new revision. Please see attachment.
Thank you again for your time.

Round 2
Reviewer 2 Report
Thank you for your clarifications and for the modifications made.
I do not have any further comments and the paper may be accepted in the present form with the alterations.